# In Vitro Evaluation of Lignin-Containing Nanocellulose

**DOI:** 10.3390/ma13153365

**Published:** 2020-07-29

**Authors:** Donguk Kim, Jaehyeon Jeong, Ji-Ae Ryu, Sa Rang Choi, Jung Myoung Lee, Heeyoun Bunch

**Affiliations:** 1Applied Biosciences, Kyungpook National University, Daegu 41566, Korea; ehddnr5000@naver.com (D.K.); jhjeong977@gmail.com (J.J.); 2Department of Wood Science & Technology, Kyungpook National University, Daegu 41566, Korea; air5030@naver.com (J.-A.R.); luvvchoi@naver.com (S.R.C.)

**Keywords:** lignocellulose, lignin-containing nanocellulose, cytotoxicity, HSP70 expression, green material

## Abstract

The increasing importance of environmental sustainability has led to the development of new materials that are environmentally friendly, functional, and cost-effective. Lignin-containing cellulose nanomaterials are a common example of these. The advantages of lignocelluloses include their renewability, sustainability, and functionality combined with molecular rigidity and enhanced hydrophobicity. In order to valorize these beneficial traits from lignin-containing nanocellulose, various approaches have been examined in industrial applications. However, the safety of these materials has not been tested or validated in humans. In this study, we tested 21 wt% lignin-containing nanocellulose (L-MFC) in vitro using the human lung and kidney cell lines, H460 and HEK293 cells, respectively. The cytotoxicity of cellulose, L-MFC, and lignin was compared using the water-soluble tetrazolium salt assays. In addition, the gene expressions of *HSP70* and *HSP90* as cellular stress markers treated with cellulose, L-MFC, and lignin were quantified using real-time polymerase chain reaction (PCR) and Western blotting. Our data indicated little cytotoxicity for cellulose and significant cytotoxicity for lignin and a relatively low level of cytotoxicity for L-MFC, providing the lethal median concentration (LC_50_) values of L-MFC and lignin. The gene expression of *HSP70* and *HSP90* was little affected by moderate concentrations of L-MFC. Interestingly, the lignin contained in L-MFC influenced the cell viability and the gene expression of *HSP70* and *HSP90* less than the same amount of lignin alone. These results indicate that L-MFC displays cell-type-dependent sensitivity and suggest that L-MFC could serve as a new eco-friendly material that is relatively safe for humans.

## 1. Introduction

Lignocellulose is the most abundant biomass on earth. This bio-based material has been used in many industrial applications as a form of final products, such as building materials, pulp, paper, tissue, packaging, and textile [1,2]. Recently, this renewable and biodegradable material is being further utilized as a nanocellulose material due to its novel properties, along with its environmental sustainability [3,4,5]. In addition to the unique properties of nanocellulose are coming from cellulose, the new properties of nanocellulose are mainly related with the nano-size of the materials depending on the production platform [6,7]. These novel functional properties have led to research on the production of lignin-containing nanocellulose since the pulp made of a higher residual lignin moiety could successfully produce a nanocellulose containing a higher residual lignin without extensive use of chemicals and energy consumption during the process [8,9,10,11]. In addition, this lignin-containing nanocellulose (L-MFC) could be used as a reinforcement in a polymer matrix with an enhanced compatibility to petroleum-based polymers, allowing better barrier properties of the composite material [12,13]. The use of nanocellulose has been explored in diverse applications, such as paper, paperboard, packaging, medical products, paints, and food additives [14,15]. 

Lignin-containing nanocellulose contains aromatic and carbohydrate polymers, such as lignin, hemicellulose, and cellulose, respectively [16]. Toxicity has been debatable for cellulose and lignin [17,18,19]. Cellulose is a hydrophilic material with a good ability to absorb water, which is important for its utilization as tissue and hygiene products. On the other hand, lignin is insoluble in water and has little ability to absorb water. For example, lignin-removed hulls can absorb 53% more water than non-processed hulls [20]. Cellulose is an abundant biomass, safe for humans and recyclable, making it eco-friendly. In addition, it is both rigid and flexible, which allows it to be developed as a substitute for plastics, such as polyethylene [21]. Microfibrillated cellulose, in particular, is a promising new material and has been processed into biodegradable, food-grade bags and packages for commerce [4].

In addition to the properties of cellulose that makes it an ideal alternative to chemical polymers, lignin-containing nanocellulose confers additional advantages, including cost-effectiveness for customers and industrial uses and reduced process of refining in manufacturing [22]. In addition, it is projected that L-MFC may have an enhanced hydrophobicity, stability, and rigidity, which could enhance the water-repellency, endurance, and neutrality of the products manufactured from it. However, its toxicity must be determined to ensure its safety for humans both in vitro and in vivo. In vivo analyses are performed on animals, such as mice and rabbits, and in vitro studies use cell lines. The strength of in vivo studies is that the effects of and physiological roles played by a substance can be analyzed in a living organism. However, such animals might have different responses to a substance than humans. In vitro analyses have fundamental limitations, including whether the cell lines derived from human tissues properly represent cells in a living body. Despite their limitations, the strength of in vitro studies is that they can provide controlled experimental conditions that allow more precise and sensitive measurements, leading to more reproducible results. Both in vivo and in vitro studies are essential for understanding the effects of a compound and to validate its safety to humans. 

We have produced lignin-free nanocellulose (cellulose), 21 wt% lignin-containing cellulose (L-MFC), and pure lignin for this study. The cellulose and L-MFC were prepared by a high-pressure homogenizer from a commercial bleached pulp and an in-house organosolv pulp, respectively. Lignin was precipitated by acidifying with sulfuric acid from the organosolv black liquor. To evaluate the cytotoxicity of L-MFC, we performed a water-soluble tetrazolium salt (WST) assay using human kidney and lung cell lines, HEK293 and H460, respectively [23,24]. The cytotoxicity of L-MFC was compared to those of lignin and cellulose, and the susceptibility and LC_50_ values of L-MFC for the cells were monitored and measured. In addition, we evaluated the cellular stresses in L-MFC-treated cells by monitoring the representative heat-shock genes, *HSP70* and *HSP90* [25,26] in HEK293 and H460 cells. *HSP70* is a stress-inducible gene, and the expressed protein functions as a chaperone protein to facilitate the folding of nascent polypeptides and misfolded proteins [27]. A range of stresses can induce *HSP70* transcription, including osmotic pressure, hypoxia, hyperthermia, and chemical and physical pressure [25,28]. HSP90 is also a chaperone protein for a number of essential proteins in diverse cellular pathways [29]. The gene expressions of *HSP70* and *HSP90* in ribonucleic acid (RNA) and protein level were measured with real-time PCR and Western blotting, respectively.

## 2. Materials and Methods 

**Nanocellulose and lignin preparations.** A commercial hardwood bleached kraft pulp (Moorim P&P Co., Ltd, Ulsan, Korea) and radiata pine chip (Sunchang Industrial Co., Ltd, Santiago, Chile) have been used as feedstocks for preparations of lignin-free and lignin-containing nanocelluloses, respectively. For the production of lignin-free nanocellulose (cellulose), the bleached kraft pulp was refined to a 100 mL Canadian Standard Freeness (CSF) in a laboratory Valley beater (DM-822, Daeil Machinery Co., Daejeon, Korea) and then was subjected to a 3 passes of a high-pressure homogenization process (Panda Plus 2000, GEA Niro Soavi, Parma, Italy) at a pressure range from 80 MPa to 110 MPa at a flow rate of 10 liters/hour (L/h). In order to produce the lignin-containing nanocellulose, the radiata pine chip of a dimension of 30(H) × 5(W) × 5(T) mm was cooked by organosolv pulping process; the pulping was carried out with a cooking wood/liquor ratio of 1:2 (w/w) using a mixture of glycol ether/sulfuric acid (97:3 (v/v)) at 120 °C for 120 min in an autoclave (HST 506-6, Hanbaek ST Co., Bucheon, Korea). The pulp was filtered using 0.5 N NaOH (Daejung Chemicals, Seoul, Korea) and washed with distilled water ((Daejung Chemicals, Seoul, Korea) to be neutralized. The lignin-containing organosolv pulp was refined and then homogenized to produce the lignin-containing nanocellulose with the same production condition mentioned above. The black liquor collected from the organosolv pulping was acidified (ca. pH 3) by a drop-wise addition of a concentrated sulfuric acid (Daejung Chemicals, Seoul, Korea) to collect the isolated lignin from the black liquor. The collected lignin was carried out Klason lignin, which intended to remove the small portion of sugars associated with the lignin during the precipitated process in order to produce a sugar-free lignin. All samples were neutralized with an excess washing with distilled water. The lignin-free and lignin-containing nanocelluloses were solvent-exchanged with tert-butyl alcohol and then subjected to a freeze drying. The isolated lignin and the two types of nanocelluloses were vacuum-dried at 55 °C for at least 3 days until further use.

**Characteristics of nanocelluloses and lignin.** Klason lignin and acid soluble lignin were determined by TAPPI T 222 and TAPPI UM 250, respectively, and the sum of Klason lignin and acid soluble lignin was expressed as lignin contents in the samples. The neutral sugars in the samples were determined by gas chromatography (HP-6890, Agilent, Santa Clara, CA, USA) fitted with a capillary column (SP-2330, Supelco, Bellefonte, PA, USA) of the alditol acetate method [30]. All chemical assays were carried out in triplicate. In order to determine morphological characteristics of the cellulose and L-MFC using an optical microscopy (BX 50, Olympus Optical Co., Ltd., Tokyo, Japan), 10 glass slide samples of the materials were prepared by dropping 20 µL of diluted sample suspension (less than 0.0001%) onto 10 different spots on the cleaned glass slide. The slide glass was thoroughly cleaned by immersing the slide glass in a mixture of ethanol (Daejung Chemicals, Seoul, Korea) and 1N HCl (1:1, v/v, Daejung Chemicals, Seoul, Korea) for overnight, followed by ultrasonic cleaning in deionized water three times. The glass slide with the cellulose and L-MFC sample attached was conditioned in a 50% relative humidity (RH) desiccator at room temperature for at least 1 week. At least 100 images were taken at different magnification by the optical microscopy (BX 50, Olympus Optical Co., Ltd., Tokyo, Japan) and were analyzed with the I-solution software (IMT I-Solution Inc., Vancover, Canada). Scanning electron microscopy (SEM, JSM-7900F, JEOL Ltd., Tokyo, Japan) images of the samples were obtained at an accelerating voltage of 10 kV. 

**Cell culture and experimental conditions.** HEK293 (Human embryonic kidney 293, ATCC, Manassas, VA, USA) and H460 cells were grown in complete media, composed of DMEM (Dulbecco’s modified eagle medium, Corning, Corning, NY, USA) supplemented with 10% FBS (Fetal bovine serum, Gibco, Gaithersburg, MD, USA) and 1% penicillin/streptomycin (P/S) solution (Gibco, Gaithersburg, MD, USA). The H460 cells were gifted by Dr. Dong-Hyung Cho’s laboratory in the Department of Life Sciences at Kyungpook National University. The cells were split at approximately 80% confluence, and the media were exchanged in every 3–4 days. 

**Cytotoxicity tests.** Cells were split into 96 well plates. Approximately 4 × 10^4^ cells were seeded in each well and cellulose, L-MFC, and lignin were added, according to the targeted concentrations. These chemicals were dissolved in Milli-Q ultrapure water (Merck, Burlington, MA, USA), and the solution was added to the cell media as 20% (v/v, 30 μL chemical solution to 120 μL cell culture media). After 24–48 h incubation, 10% (v/v) water soluble tetrazolium salt (WST, DoGen, Inc., Seoul, Korea) was added to each well, following the manufacturer’s instructions. The development of an orange color was monitored in the enzyme-substrate reaction using spectrophotometry at 450 nm (Tecan Sunrise, Männedorf, Switzerland).

**Real-time PCR.** HEK293 and H460 cells were grown to 60–70% confluence in six well plates and were replaced with the fresh media before applying cellulose, L-MFC, and lignin were applied to the target concentrations of 20% (v/v, 400 μL chemical solution to 1600 μL cell culture media). After 24 h incubation, the cells were washed with cold PBS (Phosphate-buffered saline, Gibco, Gaithersburg, MD, USA) once and scraped. The cells were washed again with cold PBS twice before the total RNA molecules were extracted using the Qiagen RNeasy kit (Qiagen, Hilden, Germany). cDNAs were constructed from 136–600 ng of the collected RNAs using ReverTra Ace qPCR RT Master Mix (Toyobo, Osaka, Japan). cDNA was analyzed through qPCR using SYBR Green Realtime PCR Master Mix (Toyobo, Osaka, Japan), according to the manufacturer’s instructions (QuantStudioTM 3 Real-Time PCR System, Applied Biosystems, ThermoFisher, Waltham, MA, USA). The thermal cycle used was 95 °C for 1 min as pre-denaturation, followed by 45 cycles of 95 °C for 15 s, 55 °C for 15 s, and 72 °C for 45 s. The primers used for the experiments were purchased from Integrated DNA Technology (Coralville, IA, USA) and are summarized in Table 1. 

**Western blotting.** HEK293 and H460 cells were grown to 60–80% confluence in six-well plates and were treated with the chemicals in the same manner as described above. The cells were washed with cold PBS twice and scraped in RIPA buffer (Cell Signaling, Cat. 9806, Danvers, MA, USA). The protein concentration of each sample was measured using a Bradford assay with Bio-Rad Protein Assay Dye Reagent Concentrate (Bio-Rad, Hercules, MA, USA) and via spectrophotometry at 595 nm (Tecan Sunrise^TM^ Absorbance Microplate Reader, Männedorf, Switzerland). From the measured protein concentration, a total of 7 μg proteins per sample was loaded on 7% sodium dodecyl sulfate-polyacrylamide gels, blotted onto a nitrocellulose membrane, and probed for HSP70 and β-ACTIN using a corresponding antibody (HSP70, Santa Cruz sc-32239, Dallas, TX, USA; β-ACTIN, Invitrogen MA5-15739, Carlsbad, CA, USA) in Western blotting assays. 

**Statistical analyses.** One-way ANOVA, followed by a post-hoc Tukey’s honest significant difference (HSD) test, were used to determine differences in toxicity percentages (*p* < 0.05) (SAS for Window release 6, SAS Institute, Cary, NC, USA). Log-probit regression was used to determine the LC_50_ based on corrected mortality from different chemical concentrations (SAS for Windows release 6, SAS Institute, Cary, NC, USA). All analyses were performed in SAS version 9.4. All of the graphs were drawn in Prism 8 (GraphPad, Inc., San Diego, CA, USA).

## 3. Results

Three different types of materials from a commercial hardwood kraft pulp and radiata pine were prepared by different production routes. The chemical compositions of the prepared cellulose, L-MFC, and lignin are presented in Table 2. The morphological characteristics of the nanocelluloses are shown in Figure 1A,B.

Our data showed that the chemical compositions of the samples were significantly different, especially in residual lignin content, due to their feedstocks and the applied chemicals during the production platform (Table 2). Compared to the nanocellulose from the bleached hardwood kraft pulp, L-MFC had about 21 wt% residual lignin with a slightly higher mannose content which is the typical main heterosaccharide of softwood (Table 2). The results showed a successful preparation of lignin-rich nanocellulose.

The nanocellulose had a higher xylose content compared to L-MFC since the bleached hardwood kraft pulp was used as a feedstock [31]. The lignin content in the isolated lignin was >99.9% because the original lignin content of the precipitated lignin from the black liquor was 92.7%, but the precipitated lignin was further hydrolyzed with an additional Klason lignin process to purify the carbohydrates associated with the residual lignin in the precipitated lignin with concentrated sulfuric acid.

The SEM images of the prepared cellulose and L-MFC are shown in Figure 1A. The results of the particle size analysis of the two different nanocelluloses measured from the optical images are shown in Figure 1B. The average length of the lignin-free nanocellulose and L-MFC was 153.2 µm ± 116.3 µm and 206.5 µm ± 167.7 µm, respectively. In addition, the average diameter was 8.2 µm ± 3.8 µm and 11.8 µm ± 4.1 µm for the cellulose and L-MFC, respectively. Based on the average length and diameter from the optical images, the aspect ratio (L/d) was calculated to be 19.2 and 22.5 for L-MFC and cellulose, respectively, suggesting that lignin-free nanocellulose had more slender particles than L-MFC, as shown in Figure 1B. Although the dimensions of nanocellulose and L-MFC prepared might be slightly coarser, we determined to use these samples for the cytotoxicity analyses without any further preparation.

Next, we prompted to evaluate the prepared nanocellulose, L-MFC, and pure lignin for the effect on the proliferation of human cells. When these chemicals are used in commercial products, kidney and lung are the most likely organs to be affected, through ingestion and inhalation, respectively. Therefore, their safety for these organs and their cells needs to be validated.

First, we tested and compared the cytotoxicity of L-MFC, cellulose, and lignin in human kidney cell line, HEK293 cells. Each of the chemicals was applied to the cells at varying concentrations for 24 h before WST analyses. Cellulose that was applied at concentrations of 0.1% or 0.2% barely affected cell viability, relative to controls (*p* > 0.1, Figure 2A). On the other hand, lignin showed significant toxicity when applied at concentrations of 0.02%, 0.05%, 0.075%, 0.1%, and 0.15% (Figure 2B). More than 90% of cells were dead in the media containing a lignin concentration as low as 0.02% (*p* < 0.0001, Figure 2B). L-MFC displayed a relatively modest cytotoxicity at the concentrations of 0.025% and 0.05%; yet, at higher concentrations of 0.1% and 0.2%, cell viability dropped by 60% to 85%, respectively, relative to the water control (*p* < 0.0001, Figure 2C). The lignin content in L-MFC was 21% of the solid, so the percentage of lignin in L-MFC was calculated to compare the cytotoxicity of lignin in the pure compound and L-MFC. Interestingly, the results showed that the lignin contained in L-MFC has lower toxicity than the pure lignin form (Figure 2D), suggesting that the chemical and physical properties may differ between pure lignin and the lignin in L-MFC. The LC_50_ value of L-MFC in HEK293 cells was derived from the cytotoxicity assays using the SAS program (Table 3). It showed a relatively mild cytotoxicity with the LC_50_ value as 0.102%.

Next, the cytotoxicity of L-MFC, cellulose, and lignin was examined in human lung cell line, H460 cells. As in HEK293 cells, cellulose did not display cytotoxicity at concentrations of 0%, 0.1%, 0.2%, or 0.4% in H460 cells (Figure 3A). Lignin dramatically inhibited cell growth in a dose-dependent manner but a bit less severely in H460 cells than in HEK293 cells (Figure 3B). For example, approximately 50% of cells survived at 0.02% lignin in H460 cells, compared to 10% in HEK293 cells (Figure 2C and Figure 3B). L-MFC also exhibited a dose-dependent cytotoxicity in H460 cells that was a bit milder than that of HEK293 cells (Figure 3C). At 0.15% and 0.2%, more than 60% and 40% of cells survived (Figure 3C). We compared the cytotoxicity of lignin between pure lignin and L-MFC by calculating the lignin content. The result was consistent as in HEK293 cells in that lignin in L-MFC was milder and less toxic than that in pure lignin (Figure 3D). For example, 0.01% lignin in L-MFC barely affected cell viability, but almost 40% of H460 cells died with the same amount of pure lignin (Figure 3D). This result supports the suggestion above that the lignin might have different properties in L-MFC from those of its original form. The LC_50_ values of cellulose, L-MFC, and lignin were derived from the cytotoxicity analyses using the SAS program (Table 4). The values were 0.021% and 0.182% for lignin and L-MFC, respectively, in H460 cells. 

The long-term effects of exposure to environmental chemicals can be predictable more accurately using gene expression profiling. Recently a number of studies have reported gene regulation by small environmental chemicals, both natural and synthetic [23,24,32,33]. In a previous study, vinyl benzoate, a synthetic, hydrophobic compound with a notable cytotoxicity (LC_50_ = 0.08% in HEK293), showed a marked reduction in *HSP70* expression in both RNA and protein levels [24]. HSP70 is a chaperon protein, which is critical for responses to various stresses in the cell, including heat, osmatic, chemical, and physical stresses. We examined the effects of L-MFC on the messenger RNA (mRNA) expression of *HSP70* in HEK293. The cells were treated with 0.084% L-MFC for 24 h, and the total RNAs were extracted to generate cDNAs. *HSP70* mRNA was quantified using real-time PCR. The results showed a slight reduction, less than 20%, in the *HSP70* mRNA level in L-MFC-treated cells (Figure 4A). The protein expression of HSP70 was quantified through immunoblotting. Consistent with the real-time PCR result, 0.042% and 0.084% L-MFC slightly altered the protein level of HSP70, while it was to an extent comparable to the control (Figure 4B). The WST results indicated little cytotoxicity below 0.1% L-MFC. We interpret these data to indicate that L-MFC at or below 0.084% might not alter the cellular stress responses or cell growth. 

The protein levels for *HSP70* mRNA were examined in H460 cells. The cells were treated with 0.1% L-MFC and 0.02% lignin for 24 h. We chose 0.1% L-MFC because this is the half of the LC_50_ value in H460 and 0.02% lignin because this is equivalent to the amount of lignin contained in 0.1% L-MFC. The result showed that the level of *HSP70* mRNA was similar between the control, L-MFC, and lignin samples (Figure 4C). We also tested *HSP90*, another chaperone protein, and its mRNA level was not altered by L-MFC treatment and increased in lignin-treated cells (Figure 4C). For the protein analyses, the cells were treated in the same conditions. Then, the HSP70 protein was quantified in the control and lignin- and L-MFC-treated cells. Immunoblotting showed that HSP70 expression was increased in lignin-treated cells but unaltered in L-MFC-treated cells (Figure 4D). These results agree with the data on the HEK293 cells that L-MFC toxicity is dose-dependent and levels of L-MFC lower than 0.1% L-MFC do not significantly alter or provoke cell-stress responses or interfere with the cell growth of H460 cells. In addition, these data reinforce the proposal that lignin in L-MFC is less toxic and presumably less disturbing to cell physiology than lignin alone in the same amount.

## 4. Discussion

We evaluated the cytotoxicity of L-MFC and its effects on cell stress regulators in vitro. These measurements are important for the development of this material into a useful commodity and for providing the guidelines to set dose limits that will be safe to humans. The L-MFC used in this study included 21% lignin and was compared to pure lignin and cellulose. Our studies present the evidence that cellulose is innocuous and lignin shows toxicity toward humans (Figure 2 and Figure 3). Based on differential toxicity of cellulose and lignin, it has also been predicted that L-MFC would probably display some toxicity for the inclusion of lignin. Our results provide the first in vitro data on this topic. 

In human H460 cells, the LC_50_ values for lignin and L-MFC were 0.0215% and 0.1990%, respectively (Table 4). We also found that the lignin content in L-MFC is less toxic than the same amount of pure lignin (Figure 2D and Figure 3D), which is interesting and important. The physical interactions and bonds between lignin and cellulose could contribute to reducing interactions between L-MFC lignin and cells, or it might change the physical and chemical properties of lignin in L-MFC, such as precipitability and compactness, to be distinctive from pure lignin. This requires further mechanistic studies.

Although cellulose nanofibers have been reported to be innocuous [34], some previous studies reported the cytotoxicity of cellulose nanofibers [35,36]. Cellulose nanofiber with concentrations over 0.0156% induced cytotoxicity against human skin cells in vitro whereas it did not induce skin or eye irritation in tissue models [35]. Another study compared the toxicity of differently processed cellulose nanofibers to zebrafish, reporting dramatically differential mortalities of these products [36]. This indicates that the toxicity of cellulose is differentiated, depending on production methods and processes. Yet we note that the concentrations of cellulose nanofiber in these studies were below 0.03% [35,36]. Our in vitro data report and agree with little cytotoxicity with nanocellulose even at 0.2% and 0.4% in HEK293 and H460 cells, respectively.

Supporting the mild and low cytotoxicity of cellulose and L-MFC, the expression of the cell stress response genes *HSP70* and *HSP90* was not induced at the mRNA or protein levels in either HEK293 or H460 cells. *HSP70* is expressed in response to various stresses and stimulations, and the main control point is typically transcription [25]. Here, we showed that lignin did not alter the expression of *HSP70* mRNA but did alter that of HSP70 protein in H460 cells (Figure 4C,D). It will be of interest to examine how lignin could affect the translational regulation of *HSP70*. We stress that for L-MFC, neither HSP70 nor HSP90 was differentially expressed (Figure 4C), suggesting that, at given concentrations, L-MFC does not provoke the cell stress responses. 

We present the data that are fundamental and essential to understand in developing L-MFC into useful materials for human usages. We are in the process to test the potential of L-MFC to be manufactured into agricultural and medical supplies as eco-friendly and cost-effective substitutes to the current ones. Currently, we are also developing the process of manufacturing L-MFC for these purposes. In addition, the safety of the material will be tested and ensured in vivo using animals and specific cell lines, depending on the purpose of developed products. 

In summary, we propose that L-MFC is a new material economical, sustainable, and safe. The potential of this material as described here is valuable and could generate a versatile next generation material through sophisticated processing and manufacturing. In addition, in vivo examinations of L-MFC are needed to validate its safety and toxicity for humans in future.

## 5. Conclusions

We validated L-MFC for cytotoxicity and cell stress gene expression in human cell lines. L-MFC has a mild toxicity, comparable to methyl benzoate, a natural organic compound used in foods and cosmetics. Lignin in L-MFC displays a reduced toxicity in comparison to pure lignin, suggesting its different physical and chemical properties in the two forms. The expression of cell stress genes of *HSP70* and *HSP90* in human cell lines of HEK293 and H460 cells are not affected by L-MFC. We suggest that L-MFC is a relatively safe material to be developed for human uses.

## Figures and Tables

**Figure 1 materials-13-03365-f001:**
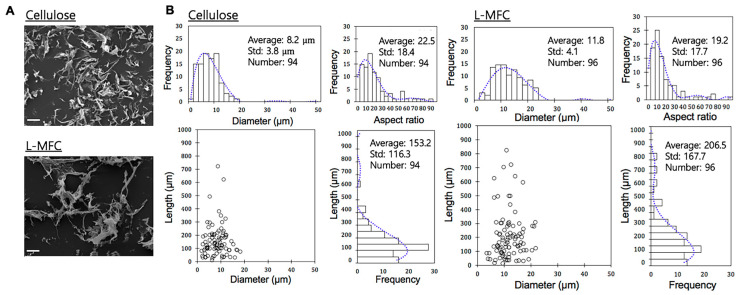
Chemical compositions and morphological characterization of cellulose, L-MFC, and lignin used in this study. (**A**) The scanning electron microscopy (SEM) images of cellulose (left) and L-MCF (right). A scale bar, 20 μm. (**B**) Optical image analyses of cellulose and L-MFC.

**Figure 2 materials-13-03365-f002:**
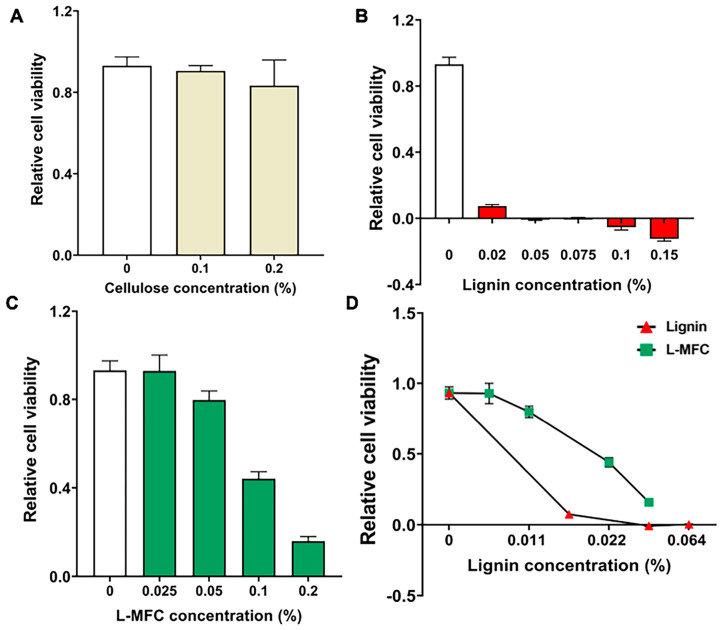
Toxicity of cellulose, L-MFC, and lignin in HEK293 cells. (**A**) Water-soluble tetrazolium salt (WST) assay results with cellulose at concentrations 0, 0.1 and 0.2%. Error bars show standard deviation (s. d.; n = 12 biological replicates). (**B**) WST assay results with lignin in increasing amounts showing marked cytotoxicity. Error bars show s. d. (n = 12). (**C**) WST assay results with L-MFC. Error bars show s. d. (n = 12). (**D**) Comparison between pure lignin and lignin contained in L-MFC for cytotoxicity. Error bars show s. d. (n = 3).

**Figure 3 materials-13-03365-f003:**
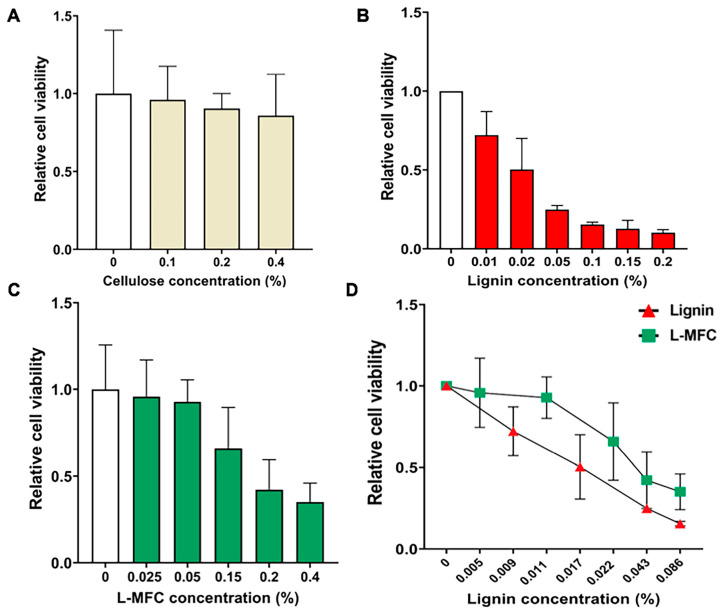
Toxicity of cellulose, L-MFC, and lignin in H460 cells. (**A**) WST assay results with cellulose at concentrations 0, 0.1, 0.2, and 0.4%. Error bars show s. d. (n = 12 biological replicates). (**B**) WST assay results with lignin in increasing amounts. Error bars show s. d. (n = 12). (**C**) WST assay results with L-MFC. Error bars show s. d. (n = 12). (**D**) Comparison between pure lignin and lignin contained in L-MFC for cytotoxicity. Error bars show s. d. (n = 3).

**Figure 4 materials-13-03365-f004:**
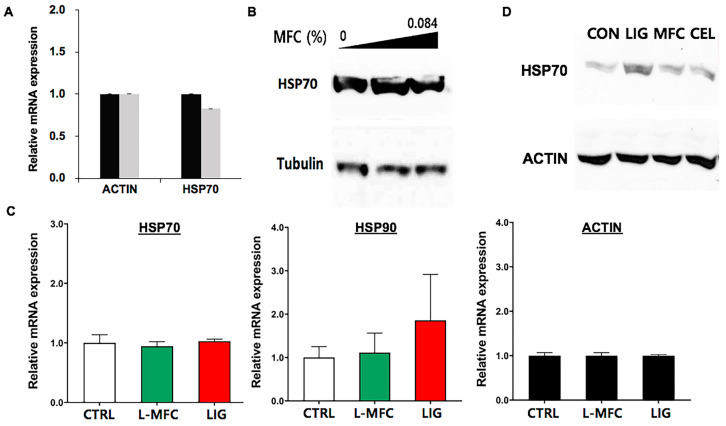
Analysis of the expression of Heat shock proteins, HSP70 and HSP90 in cellulose-, L-MFC-, and lignin-treated HEK293 and H460 cells. (**A**) qRT-PCR results showing *HSP70* mRNA expression in L-MFC-treated HEK293 cells. Black bars, untreated control, gray bars, L-MFC-treated cells. Error bars, s. d. (n = 3). β-ACTIN was used as a reference gene in all qRT-PCR analyses in this study. (**B**) Immunoblotting results probing HSP70 in L-MFC-treated HEK293 cells. Tubulin, a loading and reference control. (**C**) qRT-PCR results probing *HSP70* (left), *HSP90* (middle), and ACTIN (right, reference gene) mRNA in L-MFC- or pure lignin-treated H460 cells. CTRL, negative control; LIG, pure lignin. Error bars, s. d. (n = 3). (**D**) Western blotting results probing HSP70 in cellulose, L-MFC-, and lignin-treated H460 cells. CON, negative control; Lig, pure lignin; MFC, L-MFC; CEL, cellulose.

**Table 1 materials-13-03365-t001:** Primer sequences used for qRT-PCR in this study.

Gene	Primer Sequence
HSP70 Forward	5′-ATG TCG GTG GTG GGC ATA GA-3′
HSP70 Reverse	5′-CAC AGC GAC GTA GCA GCT CT-3′
HSP90 Forward	5′-CCG TTT CTG AGA AGC AGG GCA-3′
HSP90 Reverse	5′-CTG TCT GAA GGC CAG TGA CG-3′
ACTIN Forward	5′-GCC GAC AGG ATG CAG AAG GAG ATC A-3′
ACTIN Reverse	5′-AAG CAT TTG CGG TGG ACG ATG GA-3′

**Table 2 materials-13-03365-t002:** Chemical compositions of cellulose, lignin-containing nanocellulose (L-MFC), and pure lignin.

Samples	Lignin (%)	Total Sugar (%)	Relative Neutral Sugar Composition (%)				
			Glucose	Mannose	Galactose	Xylose	Arabinose
Cellulose	0.0 ± 0.0	96.9 ± 0.1	84.2	1.4	0	13.2	1.2
L-MFC	21.2 ± 0.6	73.1 ± 0.1	92.3	4.7	3	0	0
Lignin	>99.9 ± 0.1	<0.1 ± 0.1	<0.1	n/d	n/d	n/d	n/d

n/d, not determined.

**Table 3 materials-13-03365-t003:** LC_50_ value of L-MFC in HEK293 cells.

Cell	Treatment	LC_50_ (%)	95% CI (Lower–Upper)	Slope ± SEM	χ^2^ (df)
HEK293	Lignin	n/d	n/d	n/d	n/d
	L-MFC	0.102	0.092–0.113	3.5704 ± 0.3065	1.93 (2)

Cl, confidence interval; SEM, standard error of mean; n/d, not determined.

**Table 4 materials-13-03365-t004:** LC_50_ value of L-MFC in H460 cells.

Cell	Treatment	LC_50_ (%)	95% CI (Lower–Upper)	Slope ± SEM	χ^2^ (df)
H460	Lignin	0.021	0.017–0.264	1.4411 ± 0.1275	2.89 (4)
	L-MFC	0.182	0.156–0.221	1.7212 ± 0.1630	5.23 (3)

Cl, confidence interval; SEM, standard error of mean; n/d, not determined.

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
