# Peer review of "In Vitro Evaluation of Lignin-Containing Nanocellulose"

_materials, 2020, doi:10.3390/ma13153365_

Round 1

Reviewer 1 Report

This paper, entitled In vitro evaluation of lignin-containing nanocellulose, is a scholarly work and can increase knocledge in this domain. The authors provide an interesting study that is relevant to Materials and should generate new knowledge. This study is in the spotlight of the currenty research work. The paper is well written and well related to existing literature. The abstract and keywords are meaningful.

I have some specific and general comments:

  • Authors should provide more details about the choice of such feedstocks used in this stud. What is the reason(s) to select such materials? What are the amounts of such feedstocks and the potential recovery using such methodology?
  • Authors could consider the work of Acha et al. 2019. Synthesis of Nanolignin Following Ozonation of Lignocellulosic Biomass. Nanotechnol Adv Mater Sci. 2(4): 1-3.
  • About Figure 1, the Table should be inserted in the text ad not in this figure. Please provide standard deviation of data.
  • The graphs in Figure 1 are too small and should be resized in order to increase the readingness.
  • Same comment about the Table in the Figures 2 and 3.
  • Please provide a costs analysis of such methodology.
  • What are the next steps and future works? Please give some perspectives of this work.

As it, the paper is not fully acceptable for publication and requires minor revision and amendment. I encourage the authors to take into account all the comments and requests of amendments when resubmitting their revised manuscript.

I recommend the following decision: ACCEPT AFTER MINOR REVISION.

Author Response

Reviewer 1

This paper, entitled In vitro evaluation of lignin-containing nanocellulose, is a scholarly work and can increase knowledge in this domain. The authors provide an interesting study that is relevant to Materials and should generate new knowledge. This study is in the spotlight of the current research work. The paper is well written and well related to existing literature. The abstract and keywords are meaningful.

--> We thank Reviewer 1 for the positive comments and recognition of the value of the study. We have addressed and revised the manuscript according to the Reviewers’ suggestions.

I have some specific and general comments:

Authors should provide more details about the choice of such feedstocks used in this study. What is the reason(s) to select such materials? What are the amounts of such feedstocks and the potential recovery using such methodology?

--> We used three materials including lignin-free nanocellulose, lignin-containing nanocellulose, and isolated lignin from two feedstocks of hardwood and softwood to evaluate toxicity of each material for further utilization in the area of food packaging to replace the petro-based polymers. For the preparation of lignin-free nanocellulose, we used a commercial bleached pulp with a series of mechanical grinding. In this preparation, ca. 99% of the bleached pulp could be recovered as the lignin-free nanocellulose. If we considered that the typical yields of bleached hardwood pulp from hardwood by kraft pulping process and the lignin-free nanocellulose preparation done by this work would be around 48% and 99%, respectively, the yield calculated from the lignin-free nanocellulose from hardwood would be around 47.5%. For the preparation of lignin-containing nanocellulose, we used a pine with organosolv pulping used in this work, producing approximately 65% yield. Thus ca. 64.5% of lignin-containing nanocellulose was recovered from the pine wood. In addition, the yield of precipitated lignin from the black liquor was around 17.5% with the method we used.

Authors could consider the work of Acha et al. 2019. Synthesis of Nanolignin Following Ozonation of Lignocellulosic Biomass. Nanotechnol Adv Mater Sci. 2(4): 1-3.

--> We thank Reviewer 1 for suggesting the reference regarding to applications of lignin nanoparticle in potential industrial raw materials. We added it into the references.

About Figure 1, the Table should be inserted in the text ad not in this figure. Please provide standard deviation of data.

--> We have moved all the tables in figures to the text, following Reviewer 1’s suggestion. The standard deviation was added to each sample in the revised manuscript.

The graphs in Figure 1 are too small and should be resized in order to increase the readingness.

--> We have resized graphs in Figure 1 to be as large as possible in the revised manuscript. 

Same comment about the Table in the Figures 2 and 3.

--> We appreciate Reviewer for this suggestion once more.  The Tables in figures have been moved to the text as Table 2–4 in the revised manuscript. 

Please provide a costs analysis of such methodology.

--> As a matter of fact, we are still thoroughly investigating the organosolv pulping process used in this study to get an optimized production route with a simulation tool. As mentioned earlier, the total recovery of materials including lignin-containing nanocellulose and isolated lignin from barked pine wood was around 82%. This value indicates that we are still missing 18% of the feedstock and trying to gather the information as much as possible with various analytical instrumentations. These results should be published in near future with materials balance and cost analysis to valorize the woody biomass.

What are the next steps and future works? Please give some perspectives of this work.

--> We appreciate Reviewer 1 for this inquiry. The next steps and future works have been incorporated in the discussion section of revised manuscript. 

As it, the paper is not fully acceptable for publication and requires minor revision and amendment. I encourage the authors to take into account all the comments and requests of amendments when resubmitting their revised manuscript.

I recommend the following decision: ACCEPT AFTER MINOR REVISION

--> We have amended the manuscript according to all the comments and requests by Reviewer 1 in the revised one. We thank Reviewer 1 for his or her constructive comments and suggestions that helped to improve the manuscript.

Reviewer 2 Report

The topic is interesting and important for the bio-based nanomaterial community. However, I feel that there are some errors and lack of knowledge related to nano-scale cellulose and lignin. Therefore I suggest that the authors revisit the literature in this field.

Line 38: In addition to unique properties coming from cellulose, the new properties are nano-specific i.e. due to the nano-size of the material.

Line 49: There are publications in which toxicity has been debated for cellulose also. e.g. Cullen et al. 2002. Tumorigenicity of cellulose fibers injected into the rat peritoneal cavity. Inhalation Toxicology 14(7), 685-703.

Line 50: Cellulose is a hydrophilic material and has a good ability to absorb water, thus their utilization in tissue and hygiene products.

Line 62: the reference to pesticides is rather far-fetched and the context is unclear. It may distract the readers (it certainly did so to me). It would suffice to state that the safety (or toxicity) of any new material must be tested. The explanation between in vivo and in vitro studies was good though.

Line 79-91: Presentation of results does not belong to introduction

Line 114: Characterisation of nanocelluloses was rather superficial. Laser diffraction measurement generally has not a good response with fibril-shaped materials.

Figure 1. The placing of the Figure 1 is strange as it is among the Materials and Methods with no prior reference to the Figure nor any explanation of the results.

Line 181: Referring again to the superficial characterisation. It is impossible to distinguish nano-scale materials from the SEM images with such low resolution. Details of the image analysis should be added: how many fibrils were measured from each image, how many images were analysed per sample? According to the measurements, the materials were hardly nanocellulose, as their dimension were in the microscale. In reality, there might be nano-scale material as well but these are not detected with low resolution SEM imaging. The average width of nanocellulose is <100 nm.

Line 216: Lignin is often precipitated from the spent liquor as nanoparticles. This might be one reason for their higher toxicity.

Line 245: Again, I do not understand the reference to synthetic environmental chemicals.

Introduction / Conclusions: No earlier published results on nanocellulose toxicity are presented nor compared to the results obtained here.

Author Response

Reviewer 2

The topic is interesting and important for the bio-based nanomaterial community. However, I feel that there are some errors and lack of knowledge related to nano-scale cellulose and lignin. Therefore I suggest that the authors revisit the literature in this field.

--> We thank Reviewer 2 for the positive comments.  We have revised the manuscript according to the Reviewer’s suggestions.

Line 38: In addition to unique properties coming from cellulose, the new properties are nano-specific i.e. due to the nano-size of the material.

--> We thank the reviewer’s comments. The sentence has been changed and revised the manuscript according to Reviewer’s the valuable suggestions.

We revised the manuscript: In addition to unique properties coming form cellulose, the new properties of nanocellulose are mainly related with the nano-size of the materials depending on production platforms.

Line 49: There are publications in which toxicity has been debated for cellulose also. e.g. Cullen et al. 2002. Tumorigenicity of cellulose fibers injected into the rat peritoneal cavity. Inhalation Toxicology 14(7), 685-703.

--> We appreciate Reviewer 2 for the reference suggestion. The sentence has been changed to indicate the potential toxicity of cellulose and the given reference has been included in the revised manuscript.

Line 50: Cellulose is a hydrophilic material and has a good ability to absorb water, thus their utilization in tissue and hygiene products.

--> We thank Reviewer 2 for this correction. The mistake has been corrected in the text of the revised manuscript.

Line 62: the reference to pesticides is rather far-fetched and the context is unclear. It may distract the readers (it certainly did so to me). It would suffice to state that the safety (or toxicity) of any new material must be tested. The explanation between in vivo and in vitro studies was good though.

--> We appreciate this suggestion. We originally addressed the cases of natural pesticides because they are easily recognized as safe for their sources and however, they can be as harmful as synthetic ones. This simply suggests that any compounds, regardless of the origin, need to be validated for safety and safe limits for humans. According to Reviewer 2’s suggestion, the mentioned part about natural pesticides has been removed, only leaving the comparison of in vivo and in vitro studies.

Line 79-91: Presentation of results does not belong to introduction

--> The results have been removed from the introduction according to Reviewer’s suggestion.

Line 114: Characterisation of nanocelluloses was rather superficial. Laser diffraction measurement generally has not a good response with fibril-shaped materials.

--> We totally agree with the Reviewer’s comments. As the reviewer mentioned, the Laser diffraction measurement is not suitable with rod-like or fibril-shaped materials. The reason we used the technique was to identify the quality assurance during the micronization/homogenization as a convenient method. The laser diffraction measurement did not tell the exact dimensions of fibrils but we can easily identify the size reduction and particle size distribution. The average PSDs of the two samples in water are distinctly different; the cellulose has a 22.6 microns with no smaller particle portion (below 1<microns) and the L-MFC has a 42.8 microns with a smaller particle portion (below 1<microns).

At this point, we are still working on the optimal production route for the potential applications of lignin-containing cellulose depending on the nano-scale, micro-scale, and micro-fiber scale. Generally speaking, the smaller material gives a better performance and functionality due to increased surface area along with hydroxyl group. However, the production cost with high energy demand would be different aspects to be considered during the production platform. As a matter of fact, low-shear viscosity, water retention value (WRV) and strength evaluation with thin-films with a dissolution/regeneration method and a hydrogen-bonding (which is a similar method of handsheet making with vacuum assisted fabrication) from nano-/micro-fibril cellulose and/or micro-fiber sized material have been measured and characterized during the production stage. However, we did not put these values since adding these values might be out of scope in this study.

As the reviewer’s comment, we removed the laser diffraction method in the revised manuscript.

Figure 1. The placing of the Figure 1 is strange as it is among the Materials and Methods with no prior reference to the Figure nor any explanation of the results.

--> We have been replaced the Figure 1 in an appropriated position in the revised manuscript. Also revised the manuscript;

Three different types of materials from a commercial hardwood kraft pulp and radiata pine have been prepared by different production routes. The chemical compositions of the prepared cellulose, L-MFC, and lignin were presented in Table 2. The morphological characteristics of the nanocelluloses were shown in Figures 1A–1B. Our data showed that the chemical compositions of the samples were significantly different, especially in residual lignin content, due to their feedstocks and the applied chemicals during the production platform (Table 2). Compared to the nanocellulose from the bleached hardwood kraft pulp, L-MFC had about 21 wt% residual lignin with a slightly higher mannose content which is the typical main heterosaccharide of softwood (Table 2). The results showed a successful preparation of lignin-rich nanocellulose.

The nanocellulose had a higher xylose content compared to L-MFC since the bleached hardwood kraft pulp was used as a feedstock [31]. The lignin content in the isolated lignin was > 99.9% because the original lignin content of the precipitated lignin from the black liquor was 92.7% but the precipitated lignin was further hydrolyzed with an additional Klason lignin process to purify the carbohydrates associated with the residual lignin in the precipitated lignin with concentrated sulfuric acid.

The SEM images of the prepared cellulose and L-MFC are shown in Figure 1A. The results of the particle size analysis of the two different nanocelluloses measured from the optical images are shown in Figure 1B. The average length of the lignin-free nanocellulose and L-MFC was 153.2 µm ± 116.3 µm and 206.5 µm ± 167.7 µm, respectively. Also, the average diameter was 8.2 µm ± 3.8 µm and 11.8 µm ± 4.1 µm for the cellulose and L-MFC, respectively. Based on the average length and diameter from the optical images, the aspect ratio (L/d) was calculated to be 19.2 and 22.5 for L-MFC and cellulose, respectively suggesting that lignin-free nanocellulose had more slender particles than L-MFC as shown in Figure 1B. Although the dimensions of nanocellulose and L-MFC prepared might be slightly coarser, we determined to use as the samples for the cytotoxicity analyses without any further preparation.

Line 181: Referring again to the superficial characterisation. It is impossible to distinguish nano-scale materials from the SEM images with such low resolution. Details of the image analysis should be added: how many fibrils were measured from each image, how many images were analysed per sample? According to the measurements, the materials were hardly nanocellulose, as their dimension were in the microscale. In reality, there might be nano-scale material as well but these are not detected with low resolution SEM imaging. The average width of nanocellulose is <100 nm.

--> As the reviewer mentioned, the materials used in this study was not the exact nano-scale material by its scientific definition and they are materials in nano-scale, micro-scale or micro-fiber scale although we could create more nano-scale lignin-containing materials with a higher energy input /increased severity. As mentioned earlier, we are still looking for the optimal production route for potential applications of lignin-containing nano-/micro-fibril scale as well as microfiber sized scale during the production platform. In order to identify various applications with L-MFC, we have been working with two distinct approaches; 1) dissolution and regeneration process to make filaments, nano-drop and thin-film 2) hydrogen-bonding process to make thin-film, molded sheet, and porous sheet. In order to use these lignin-containing materials as a feedstock for industrial applications, we were trying to verify the toxicity, regardless of their sizes.

We measured the dimensional information with a high-resolution optical microscopy with around 100 individual fibrils with an experienced operator. The 10 slide glasses were carefully prepared with a 20 microliter of diluted sample suspension (less than 0.0001%) with 10 spots in a glass slide. The SEM images was for the general morphological information for the reader. This has been modified and revised in the manuscript:

In order to determine morphological characteristics of the cellulose and L-MFC using an optical microscopy (BX 50, Olympus optical Co., Ltd., Japan), 10 glass slide samples of the materials were prepared by dropping 20 µL of diluted sample suspension (less than 0.0001%) onto 10 different spots on the cleaned glass slide. The slide glass was thoroughly cleaned by immersing the slide glass in a mixture of ethanol and 1N HCl (1:1, v/v) for overnight, followed by ultrasonic cleaning in deionized water three times. The glass slide with the cellulose and L-MFC sample attached was conditioned in a 50% RH desiccator at room temperature for at least 1 week. At least 100 images were taken at different magnification by the optical microscopy (BX 50, Olympus optical Co., Ltd., Japan) and were analyzed with the I-solution software (IMT I-Solution Inc., Canada). Scanning electron microscopy (JSM-7900F, JEOL Ltd., Japan) images of the samples were obtained at an accelerating voltage of 10 kV.

Line 216: Lignin is often precipitated from the spent liquor as nanoparticles. This might be one reason for their higher toxicity.

--> We thank Reviewer 2 for the discussion. We agree that lignin has a distinctive physical property from cellulose or L-MFC, including the precipitability and compactness. This has been included in the discussion section of the revised manuscript as below:

We also found that the lignin content in L-MFC is less toxic than the same amount of pure lignin (Figures 2D and 3D), which is interesting and important. The physical interactions and bonds between lignin and cellulose could contribute to reducing interactions between L-MFC lignin and cells, or it might change the physical and chemical properties of lignin in L-MFC, such as precipitability and compactness, to be distinctive from pure lignin. This requires further mechanistic studies.

Line 245: Again, I do not understand the reference to synthetic environmental chemicals.

--> We assumed that Reviewer mentioned the sentences in Line 295, not Line 245 (if it is not, please excuse us and kindly point out the exact location). According to Reviewer 2’s suggestion, the reference to environmental chemicals has been removed in the revised manuscript.  

Introduction / Conclusions: No earlier published results on nanocellulose toxicity are presented nor compared to the results obtained here.

--> We appreciate Reviewer 2 for pointing this.  We have newly discussed the previous studies about nanocellulose toxicity and compared them with our present study in the Discussion section as below:

Although cellulose nanofibers have been reported to be innocuous [34], some previous studies reported the cytotoxicity of cellulose nanofibers [35, 36]. Cellulose nanofiber with concentrations over 0.0156% induced cytotoxicity against human skin cells in vitro whereas it did not induce skin or eye irritation in tissue models [35]. Another study compared the toxicity of differently processed cellulose nanofibers to zebrafish, reporting dramatically differential mortalities among these products [36]. This indicates that the toxicity of cellulose is differentiated, depending on production methods and processes. Yet we note that the concentrations of cellulose nanofiber in these studies were below 0.03% [35, 36]. Our in vitro data report and agree with little cytotoxicity with nanocellulose even at 0.2% and 0.4% in HEK293 and H460 cells, respectively.           

Round 2

Reviewer 2 Report

The authors have responded to all the comments made to the first version of the paper and therefore I can recommend the paper for publication.

However, they should still consider rephrasing sentences like on line 301 "..lignin is toxic and to human health...", perhaps to "lignin shows toxicity towards humans.." or something similar.

In addition, in their response, the authors claim several times that the process of manufacturing L-MFC is under development. Perhaps this should be stated in the paper.

Author Response

The authors have responded to all the comments made to the first version of the paper and therefore I can recommend the paper for publication.

--> We thank Reviewer 2 for the recommendation.

However, they should still consider rephrasing sentences like on line 301 "..lignin is toxic and to human health...", perhaps to "lignin shows toxicity towards humans.." or something similar.

--> We appreciate Reviewer for this suggestion.  According to Reviewer’s comment, the sentence has been revised as below:

lignin is toxic and to human health à lignin shows toxicity toward humans

In addition, in their response, the authors claim several times that the process of manufacturing L-MFC is under development. Perhaps this should be stated in the paper.

--> For Reviewer’s suggestion, a sentence has been newly incorporated in the revised manuscript (lines 326–327) as below:

Currently, we are also developing the process of manufacturing L-MFC for these purposes.
